# Developmental trajectory of voluntary alcohol consumption in adolescent mice using finite mixture modeling and Bayesian posterior probability analysis

**Nathan Yu[1], Derek Gordon [1], Hong Zou[2], Yingying Chen[3], Lei Yu [1,4]***

**1** Department of Genetics, Rutgers University, Piscataway, New Jersey, United States of America, **2** Shanghai Institute of Nutrition and Health, Chinese Academy of Sciences, Shanghai, People's Republic of China, **3** Department of Electrical and Computer Engineering, Rutgers University, Piscataway, New Jersey, United States of America, **4** Center of Alcohol & Substance Use Studies, Rutgers University, Piscataway, New Jersey, United States of America

* yu@biology.rutgers.edu

## Abstract

### Background

Alcohol use disorders (AUDs) pose a significant public health challenge, with adolescence representing a critical period of vulnerability for the initiation of alcohol consumption. Variability in drinking behaviors among individuals complicates efforts to characterize developmental trajectories, limiting our understanding of underlying biological mechanisms.

### Objective

This study aimed to identify and characterize distinct patterns of voluntary alcohol consumption in adolescent mice, using advanced statistical methods to model behavioral heterogeneity.

### Methods

Thirty-five male CD-1 outbred mice were monitored for alcohol consumption using a two-bottle free-choice paradigm from early adolescence to young adulthood (4–11 weeks of age). Finite mixture modeling, using the method implemented in the software SAS Proc Traj, was applied to categorize individual drinking behaviors into trajectory groups based on Bayesian Information Criterion (BIC) and Bayesian Posterior Probabilities (BPP).

### Results

Three distinct drinking trajectory groups were identified: non-drinkers, late drinkers, and early drinkers. Non-drinkers exhibited consistently low alcohol consumption throughout the study, late drinkers showed a significant increase in alcohol intake during adolescence-to-adulthood transition, and early drinkers maintained high levels of consumption from the start. Notably, the late and early drinkers converged on similarly high

**Data availability statement:** All relevant data are within the paper.

**Funding:** This work was supported by the National Institutes of Health of the United States (DA020555 to L.Y.). The funders did not play any role in study design, data collection and analysis, decision to publish, or preparation of the manuscript.

**Competing interests:** The authors have declared that no competing interests exist.

consumption levels by the end of the observation period. These findings highlight the heterogeneity of drinking behaviors during adolescence and its developmental implications.

## Conclusions

This study demonstrates the utility of finite mixture modeling in characterizing developmental trajectories of voluntary alcohol consumption in adolescent mice. The identification of distinct behavioral trajectory patterns provides a foundation for future investigations into the genetic, molecular, and neural mechanisms underpinning susceptibility to alcohol use disorders.

## 1. Introduction

Alcohol use disorders (AUDs) pose a significant public health challenge, contributing to substantial societal and personal costs [1–4]. Adolescence represents a pivotal developmental stage characterized by heightened susceptibility to risk-taking behaviors, including alcohol consumption [5,6]. Early exposure to alcohol during adolescence development is a critical factor of AUD development in adulthood [7–11], emphasizing the need for a better understanding of drinking behaviors in adolescence.

In this study, we investigated voluntary alcohol consumption in adolescent mice, spanning early-adolescence to young adulthood, using an outbred mouse strain to model genetic diversity similar to that in humans. By employing the statistical approach of finite mixture modeling, we categorized individual drinking trajectories. Specifically, each mouse has a set of longitudinal data, consisting of days on which alcohol consumption measurement were taken, and the value of that measurement. In finite mixture models, the trajectory curves (longitudinal data) are approximated by polynomials of order up to 4. Among the benefits of finite mixture model approaches are that they can handle missing data, unequally spaced measurements, and multiple different non-monotonic trajectory curves.

An important component of the finite mixture model is that the set of alcohol consumption curves is comprised of unique groups. As noted above, each group of similar trajectory curves (group being made up of mice with analogous curves) is described by a polynomial whose coefficients are estimated by maximum likelihood using the EM algorithm. The probability that each mouse's growth curve belongs to a group is determined by its Bayesian Posterior Probability (BPP).

With regards to the current study, this methodology allowed us to characterize the heterogeneity in drinking patterns and identify distinct developmental trajectory curves. Our findings provide valuable insights into the biological basis of adolescent alcohol consumption and establish a framework for future studies on the mechanisms underlying AUD susceptibility.

## 2. Materials and methods

### 2.1. Animals

Thirty five male CD-1 (ICR) outbred mice, three weeks old, were obtained from Shanghai Laboratory Animal Center, Chinese Academy of Sciences, Shanghai, China. Animals were housed in temperature controlled animal facilities on a 12 h:12 h light-dark cycle with food and water available ad libitum. At the beginning of the chronic alcohol drinking experiment, mice were singly housed for seven days of acclimation before the start of the alcohol drinking study. At the end of the study, all mice were euthanized by cervical dislocation. All procedures were approved by the IACUC, and were performed in accordance with the Guide for the Care

and Use of Laboratory Animals (Institute of Laboratory Animal Resources, 1996), the PRC National Standards for Laboratory Animal Quality, and the Guidelines for the Use of Experimental Animals.

## 2.2. Voluntary alcohol consumption, food and water intake measurements

Voluntary alcohol consumption in mice employing a two-bottle free-choice paradigm followed a published procedure [12]. Three weeks old mice were acclimated in single-housing cages for seven days with food and water. At four weeks of age, two-bottle free-choice procedure began with the mouse having free access to both a water-only sipper tube and an ethanol-containing sipper tube. The positions of the alcohol and water sipper tubes were counter-balanced and randomized across days. Two concentrations of alcohol solution, 5% and 10%, were used, and mice were randomly assigned to one of the alcohol solutions for the duration of the study. Mice assigned to the 5% alcohol solution regimen (Day 1–5: water only; Day 6–10: 2% alcohol; Day 11 onward: 5% alcohol) were numbered 1–18, and mice assigned to the 10% alcohol solution regimen (Day 1–5: water only; Day 6 onward: 10% alcohol) were numbered 19–35. To account for evaporation, separate tubes containing alcohol and water solutions were placed in an empty cage in the same room that housed the mice under study, and the amount of evaporation was recorded daily. Evaporated amount was subtracted from the amount of water or alcohol consumption, respectively.

## 2.3. Statistical methods

Finite mixture modeling was used to test for heterogeneity of mouse alcohol consumption values, by implementing the method Proc Traj in SAS, a procedure for group-based trajectory modeling [13,14]. This methodology has been applied to other mouse-longitudinal data studies [15]. A reliable method to determine which candidate chemotherapeutic drugs effectively inhibit tumor growth in patient-derived xenografts (PDX) in single mouse trials [15].

Inputs were mouse alcohol consumption data, specification of the number of groups (k), and specification of the order of each of the k polynomials. Outputs were coefficient estimates and the corresponding p values for each of the polynomials, estimate of the standard deviation (sigma), estimated group membership percentages with corresponding p values, BIC values corresponding to the input values, and the k Bayesian Posterior Probabilities (BPPs) for every mouse and every group. Each p value was for the t-test where the null hypothesis was that the relevant parameter was 0. Also, the BPPs were the probabilities that a given mouse is a member of group k. We called the polynomials with the estimated coefficients the trajectory polynomials, and we called each of the groups the trajectory groups.

Data based on mouse groups were evaluated by two-way analysis of variance (ANOVA) with Tukey's post hoc tests, using GraphPad Prism software (version 9.5, GraphPad Software, San Diego, CA, USA). Significance levels were noted using an alpha level of 0.05.

## 3. Results

### 3.1. Voluntary alcohol consumption in outbred mice with two-bottle free-choice paradigm

To set up a mouse model for individual patterns of voluntary alcohol consumption during adolescence, we used outbred mice, and evaluate innate tendency of voluntary alcohol consumption.

A two-bottle free-choice paradigm was employed, to measure voluntary alcohol consumption, without food or water restriction. At the beginning of the study, mice were 4-weeks old,

somewhat comparable to pre-adolescence in human development [5,6,16–19]. Except for daily weighing, no other handling or behavioral tests were performed during the voluntary alcohol consumption period, lasting 57 days, from mouse age 4 weeks to 11 weeks.

As shown in Fig 1, there was substantial variation of voluntary alcohol consumption among different mice, with some mice consuming considerably more alcohol solution than other mice.

## 3.2. Categorization of voluntary alcohol consumption by statistical approaches

**3.2.1. Heterogeneity analysis.** Given the heterogeneous nature of voluntary alcohol consumption among adolescent mice, we applied the method implemented in the method Proc Traj in SAS, a procedure for group-based trajectory modeling [13,14], to test our null hypothesis that the data comes from a homogeneous drinking group. Table 1 shows the results of testing on various quadratic models (polynomials up to order 2), to determine the most parsimonious group number.

The most parsimonious model among all groups (k) needs to satisfy the following criteria:

1. Estimated group membership probabilities are all greater than 10% with all corresponding p values less than 0.1;

2. For each of the mice, there is a greater than 95% BPP for being in one of the k groups;

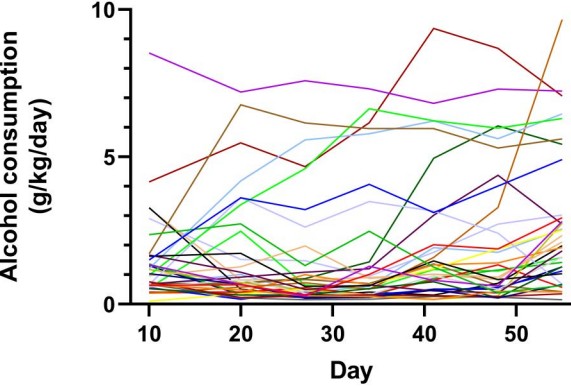

**Fig 1. Substantial variation in individual patterns of voluntary alcohol consumption in outbred mice.** Thirty five male CD-1 outbred mice at age 4 weeks, were singly housed and given two-bottle free choice of either water or alcohol solution at the start of study. Daily voluntary alcohol consumption data (averaged over 5 day period) were shown for each of the 35 mice as grams of alcohol consumed per kg body weight per day.

**Table 1. Summary analyses for determining the most parsimonious group number.**

| Number of groups (k) | All group memberships greater than 10% | P-value less than 0.1 | BPPs greater than 95% | BIC |
|---|---|---|---|---|
| 1 | N/A[a] | N/A[a] | N/A[a] | -4183.1 |
| 2 | YES | YES | YES | -3628.8 |
| 3 | YES | YES | YES | -3489.3 |
| 4 | NO | NO | YES | -3449.94 |

[a]N/A: not applicable. For k equals 1 group the group membership is 100% and there is no corresponding p value and the BPP is 1 for every mouse.

The model that meets the above criteria and has the largest BIC for all observed data is the most parsimonious model.

Our null hypothesis is that the data comes from a homogeneous drinking group (that k = 1). We reject this hypothesis if the following two conditions are met: the most parsimonious model is for k >1, and the difference (the BIC score for the most parsimonious model minus the BIC score for the model with k equals 1 group) is greater than 10 [20].

From Table 1, we observe that groups 1–3 are all candidates for being the most parsimonious model based on our criteria (1. Estimated group membership probabilities are all greater than 10% with all corresponding P values less than 0.1; and 2. For each mouse, there is a greater than 95% BPP for being in one of the k groups). When k equals 4 groups, one group membership percentage is not significantly different from 0 based on the P-value. Hence, the model of group equaling 4 is excluded from further consideration.

Studying Table 1 further, we note that k equaling three groups has the largest BIC value and it is substantially larger than the BIC values for k equaling either one or two (by a difference of 693.8 and 139.5, respectively). Based on this result, we specify that k equaling three groups is the most parsimonious model. This finding indicates that we reject the null hypothesis that there is only one group, in favor of the alternate hypothesis that mouse alcohol consumption over time for this dataset is a mixture of three groups each with their own unique trajectory. From this point forwards, we will focus our consideration of results for k equaling three groups.

**3.2.2. Different longitudinal drinking patterns.** The SAS Proc Traj results for three groups are presented in Table 2.

Table 3 shows the Bayesian Posterior Probabilities (BPPs) for all mice when k equals three groups. We note that every mouse is placed into a unique trajectory group with a high degree of certainty. This conclusion is based on the fact that every BPP is either below 0.001 or above

**Table 2. Summary of SAS Proc Traj results for three groups based on mouse alcohol consumption data.**

| The SAS System: Maximum Likelihood Estimates | | | | | |
|---|---|---|---|---|---|
| **Model: Censored Normal (CNORM)** | | | | | |
| **Group** | **Parameter** | **Estimate** | **Standard Error** | **t Statistic for H0: Parameter = 0** | **Prob> \|t\|** |
| 1 | Intercept | 1.39009 | 0.49984 | 2.781 | 0.0055 |
| | Linear | -0.05619 | 0.03587 | -1.566 | 0.1175 |
| | Quadratic | 0.00262 | 0.00056 | 4.711 | 0.0000 |
| 2 | Intercept | 1.61339 | 0.19591 | 8.235 | 0.0000 |
| | Linear | -0.061 | 0.01406 | -4.339 | 0.0000 |
| | Quadratic | 0.00109 | 0.00022 | 4.99 | 0.0000 |
| 3 | Intercept | 2.4755 | 0.44686 | 5.54 | 0.0000 |
| | Linear | 0.19721 | 0.03207 | 6.15 | 0.0000 |
| | Quadratic | -0.00225 | 0.0005 | -4.511 | 0.0000 |
| | Sigma | 1.60771 | 0.02684 | 59.898 | 0.0000 |
| Group Membership | | | | | |
| 1 | (%) | 11.42857 | 5.3944 | 2.118 | 0.0343 |
| 2 | (%) | 74.28571 | 7.4104 | 10.025 | 0.0000 |
| 3 | (%) | 14.28571 | 5.93294 | 2.408 | 0.0161 |
| BIC = -3489.34 (N=1805) | | BIC= -3465.68 (N=35) | | AIC= -3456.35 | L= -3444.35 |

**Table 3. Mouse group membership.**

| Mouse # | Group 1 Probability | Group 2 Probability | Group 3 Probability | Group Assignment |
|---------|--------------------|--------------------|--------------------|------------------|
| 1 | 1 | 0 | 0 | 1 |
| 18 | 1 | 0 | 0 | 1 |
| 29 | 0.999473 | 0.000527 | 0 | 1 |
| 31 | 1 | 0 | 0 | 1 |
| 2 | 0 | 1 | 0 | 2 |
| 3 | 0 | 1 | 0 | 2 |
| 4 | 0 | 1 | 0 | 2 |
| 5 | 0 | 1 | 0 | 2 |
| 6 | 0 | 1 | 0 | 2 |
| 8 | 0 | 1 | 0 | 2 |
| 9 | 0 | 1 | 0 | 2 |
| 10 | 0 | 1 | 0 | 2 |
| 11 | 0 | 1 | 0 | 2 |
| 12 | 0 | 1 | 0 | 2 |
| 15 | 0 | 1 | 0 | 2 |
| 16 | 0 | 1 | 0 | 2 |
| 17 | 0 | 1 | 0 | 2 |
| 19 | 0 | 1 | 0 | 2 |
| 20 | 0 | 1 | 0 | 2 |
| 21 | 0 | 1 | 0 | 2 |
| 22 | 0 | 1 | 0 | 2 |
| 23 | 0 | 1 | 0 | 2 |
| 25 | 0 | 1 | 0 | 2 |
| 26 | 0 | 1 | 0 | 2 |
| 27 | 0 | 1 | 0 | 2 |
| 28 | 0 | 1 | 0 | 2 |
| 32 | 0 | 1 | 0 | 2 |
| 33 | 0 | 1 | 0 | 2 |
| 34 | 0 | 1 | 0 | 2 |
| 35 | 0 | 1 | 0 | 2 |
| 7 | 0 | 0 | 1 | 3 |
| 13 | 0 | 0 | 1 | 3 |
| 14 | 0 | 0 | 1 | 3 |
| 24 | 0 | 0 | 1 | 3 |
| 30 | 0 | 0 | 1 | 3 |

0.999. These result indicate that three groups is a potentially parsimonious model for the mouse alcohol consumption dataset. Thus, finite mixture modeling with Proc Traj produced unequivocal results of a three-group quadratic model.

### 3.3. Patterns of developmental changes based on alcohol consumption grouping

Following the three-group membership assignments based on mouse alcohol consumption dataset, we examined the developmental patterns of alcohol consumption in these three groups. As shown in Fig 2 (mean ± SEM, n = 4 for Group 1, 26 for Group 2, and 5 for Group 3), each of the three groups displayed a distinct pattern of developmental trajectory. Group 1

mice consumed low levels of alcohol during the early part of the study (from the beginning of the study through Day 34), then increased their alcohol consumption appreciably (Day 41 and 48), until they reached their highest levels of alcohol consumption by Day 55. Group 1 mice were thus termed 'late drinkers,' to indicate the rising pattern of their alcohol consumption. Group 2 mice showed consistently low levels of alcohol consumption, and were termed 'non-drinkers' (due to the fact that the positions of the water and alcohol solution tubes were randomly assigned each day in the mouse home cage, a mouse would need to take a few licks from a solution tube to figure out which one was which; thus it would not be able to completely avoid consuming a low level of alcoholic solution). Group 3 mice started consuming high levels of alcohol as soon as alcoholic solutions were made available to them at the beginning of the study, and they sustained such high consumption levels throughout the study. Group 3 mice were therefore term 'early drinkers.'

Statistical analysis (two-way ANOVA, with Tukey's post hoc tests) indicated three phases, mostly due to the changing behavior of Group 1 ('late drinkers'), as Group 2 ('non-drinkers') and Group 3 ('early drinkers') displayed relatively consistent alcohol consumption patterns. During the early phase of the study (from the beginning of the study through Day 34), Group 1 and Group 2 mice consumed low levels of alcohol, and their alcohol consumption were statistically indistinguishable from each other during this early phase. Their low levels of alcohol consumption were both significantly different from Group 3 ('early drinkers'). During the transition phase (Day 41 and 48), alcohol consumption for each group was significantly different from any other group. Toward the end phase of the study, Group 1 and Group 3 showed similar levels of high alcohol consumption, which were significantly different from the low levels of alcohol consumption by Group 2.

To observe through the perspective of the 3-group membership assignments based on mouse alcohol consumption, we examined mouse body weight growth and feeding behaviors (mean ± SEM, n = 4 for Group 1, 26 for Group 2, and 5 for Group 3). As shown in Fig 3, the body weight of the three groups of mice were comparable throughout the duration of the study. The three groups of mice displayed comparable levels of food intake for the duration that food intake was measured (seven weeks from the beginning of the study, see Fig 4).

Liquid intake displayed a different pattern than that of food intake. As shown in Fig 5A, intake of alcoholic solution closely followed a pattern similar to that of alcohol consumption (Fig 2). Interestingly, water intake data (Fig 5B) showed that Group 3 ('high drinkers') had lower levels of water intake than the other two groups. When the total liquid intake was examined, the three groups displayed comparable levels (Fig 5C).

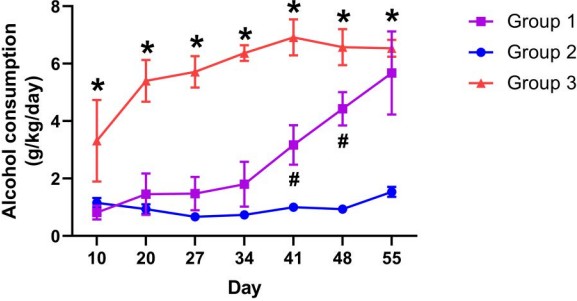

**Fig 2. Voluntary alcohol consumption of the three groups of mice.** Daily voluntary alcohol consumption data (grams of alcohol consumed per kg body weight per day, averaged over 5 day period) of the mice in each group are shown as mean ± SEM. * Significant difference from non-drinkers (Group 2). # Significant difference from both non-drinkers (Group 2) and early drinkers (Group 3).

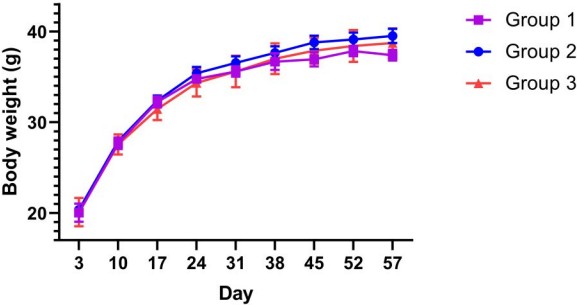

**Fig 3. Body weight of the three groups of mice.** Body weight data (g, averaged over 5 day period) of the mice in each group are shown as mean ± SEM.

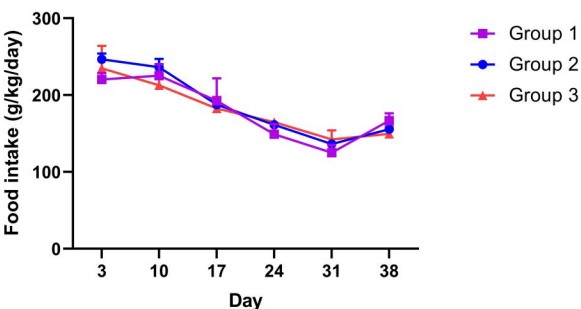

**Fig 4. Food intake of the three groups of mice.**

## 4. Discussion

Adolescence represents a critical developmental period marked by significant behavioral and neurobiological changes [5,6,21], including susceptibility to and experimenting with alcohol consumption [7–11,22,23]. Understanding the developmental trajectories of alcohol consumption during adolescence is important, as early exposure to alcohol is a known risk factor for alcohol use disorders in adulthood [24,25]. In this study, we utilized a two-bottle free-choice paradigm to assess voluntary alcohol consumption in adolescent mice. We chose to use an outbred mouse strain CD-1, which has a diverse genetic makeup comparable to human populations [26]. By studying voluntary alcohol consumption in mice starting at 4 weeks old and continuing across a 7-week time span, we covered the development period of early adolescence-to-adulthood equivalent to that in humans [5,6,16–19]. By using advanced statistical modeling, we were able to characterize distinct patterns of voluntary alcohol consumption, gaining insights into the heterogeneity of alcohol drinking behaviors in adolescent mice.

Our results showed considerable variability in alcohol consumption among individual mice, highlighting the complexity of behavioral phenotypes associated with alcohol use (Fig 1). Using the SAS Proc Traj finite mixture modeling approach [13,14], we identified three distinct trajectory groups: non-drinkers, late drinkers, and early drinkers (Table 1). The selection of the three-group model as the most parsimonious was supported by Bayesian Information Criterion (BIC) values and Bayesian Posterior Probabilities (BPPs) (Table 2), which confirmed high certainty in group membership for all mice (Table 3). These statistical methods provided an objective framework to categorize individual consumption behaviors.

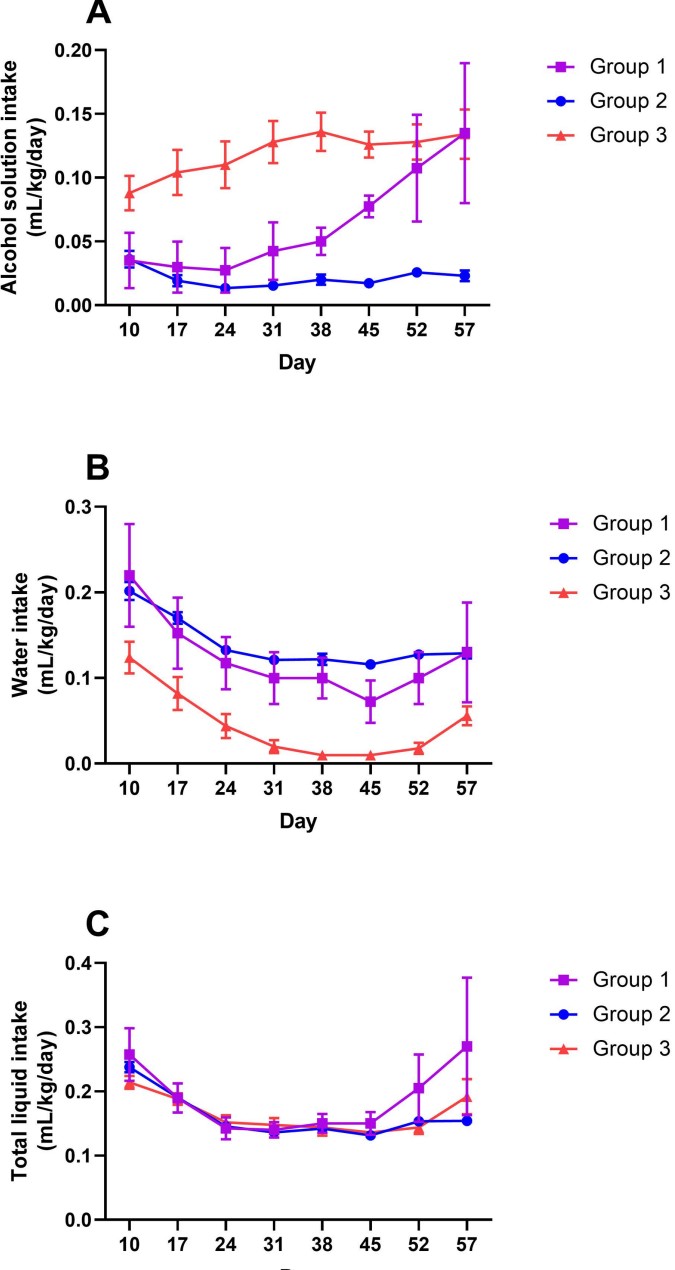

**Fig 5. Liquid intake of the three groups of mice.** Shown are mean ± SEM (n = 4 for Group 1, 26 for Group 2, and 5 for Group 3). **(A)** Daily intake of alcohol-containing solution (mL of alcohol solution consumed per kg body weight per day). **(B)** Daily water intake (mL of water consumed per kg body weight per day). **(C)** Total liquid intake (the sum of mL of alcohol solution consumed and mL of water consumed, per kg body weight per day).

Each group exhibited a unique developmental pattern of alcohol consumption (Fig 2). The non-drinkers (Group 2) displayed consistently low levels of alcohol consumption throughout the adolescence-to-adulthood developmental stage. The late drinkers (Group 1) showed low levels of alcohol consumption at the beginning of the study (Fig 2, start through study day 34), with levels comparable to those displayed by non-drinkers. As mice in Group 1 transitioned

from adolescence to young adulthood (Fig 2, study days 41–48), significantly notable difference emerges from that of non-drinkers, reflecting a marked increase in alcohol consumption, thus suggesting a delayed but robust engagement with alcohol. Toward the end of the study (Fig 2, study day 55), Group 1 mice reached high levels of alcohol consumption similar to those by the early drinkers (Group 3), which began consuming high levels of alcohol immediately upon its availability, maintaining this pattern throughout the study (Fig 2). Notably, the late drinkers and early drinkers converged on comparable high levels of consumption by the end of the study, indicating that these groups may represent distinct developmental pathways leading to high alcohol intake. As outbred mice, the high levels of alcohol consumption achieved by both Group 1 and Group 3 are comparable to several of the inbred mouse strains [27], although still below those by C57BL/6 [27–29], as C57BL/6 mice are known to show higher ethanol intake than many other mouse strains [30].

The voluntary alcohol consumption paradigm in our study compared two regimens of alcohol solution: 5% (mouse 1–18) and 10% (mouse 19–35). As shown in Table 3, it appears that there was no difference between the two regimens – half of the mice in Group 1 (late drinkers) were from the 5% regimen (mouse 1 and 18), and the other half were from the 10% regimen (mouse 29 and 31); similarly, Group 3 (early drinkers) membership was also relatively evenly distributed – three mice were from the 5% regimen (mouse 7, 13 and 14), and two were from the 10% regimen (mouse 24 and 30). This suggests that regardless of the alcohol concentration of the regimens, mice in a group consumed similar amounts of alcohol (as reported in g/kg/day, see Fig. 2).

Based on the 3-group membership assignments, the body weight of the three groups of mice were comparable throughout the duration of the study (Fig 3). The three groups of mice also displayed comparable levels of food intake for the duration that food intake was measured (seven weeks from the beginning of the study, see Fig 4). It should be pointed out that the increase of alcohol intake of Group 1 started around the time of seven weeks, which may have an effect on food intake. We did notice that, however, for the high alcohol consumption of Group 3, their food intake levels were comparable to that of the other groups during the first seven weeks of the study, despite Group 3 mice consuming more alcohol during this period.

These findings have significant implications for understanding the developmental process of alcohol consumption. The distinct patterns of drinking behavior observed here are likely influenced by the underlying genetic and neurobiological processes, and innate factors at molecular and cellular levels that modulate these processes. Future studies may help unravel the molecular mechanisms driving these distinct patterns of voluntary alcohol consumption in adolescent mice.

Our study also highlights the utility of finite mixture modeling for analyzing complex behavioral data. Specifically, this approach enabled precise categorization of individual trajectories of behavior during development, providing a way for robust and reliable behavior phenotyping. The application of such an approach extends beyond the study of alcohol consumption, offering a powerful tool that may be utilized to study other developmentally regulated behaviors in biomedical research.

In summary, the present study provides an example and a framework for categorizing developmental patterns of alcohol consumption in adolescent mice, enabling a more ordered pattern interpretation of heterogeneity in individual study subjects. The identification of distinct trajectory groups of voluntary alcohol consumption during adolescence development suggests the existence of specific molecular and neurodevelopmental mechanisms that underlie such patterns. Future studies leveraging these findings could investigate the genetic, molecular, and neural correlates of drinking patterns, contributing to a better understanding of the biological mechanisms associated with alcohol use disorders.

## Acknowledgements

The authors would like to thank Qinglian Xie for technical assistance.

## Author contributions

**Conceptualization:** Derek Gordon, Hong Zou, Lei Yu.

**Data curation:** Nathan Yu, Hong Zou.

**Formal analysis:** Nathan Yu, Derek Gordon, Yingying Chen.

**Funding acquisition:** Lei Yu.

**Investigation:** Nathan Yu, Derek Gordon, Hong Zou.

**Project administration:** Lei Yu.

**Resources:** Hong Zou, Lei Yu.

**Software:** Derek Gordon.

**Supervision:** Derek Gordon, Hong Zou, Lei Yu.

**Visualization:** Nathan Yu, Lei Yu.

**Writing – original draft:** Nathan Yu, Derek Gordon, Lei Yu.

**Writing – review & editing:** Nathan Yu, Derek Gordon, Hong Zou, Yingying Chen, Lei Yu.

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
