## [Decision Letter · Decision Letter 0]

21 Feb 2025

PONE-D-24-56621Developmental trajectory of voluntary alcohol consumption in adolescent mice using finite mixture  modeling and Bayesian posterior probability analysisPLOS ONE

Dear Dr. Yu,

Thank you for submitting your manuscript to PLOS ONE. After careful consideration, we feel that it has merit but does not fully meet PLOS ONE’s publication criteria as it currently stands. Therefore, we invite you to submit a revised version of the manuscript that addresses the points raised during the review process.

 Reviewer #1 and I find your work to be a valuable contribution to the field of alcohol research. You should know that I took on the role of Reviewer #2 due to delays in the peer-review process. Reviewer #1 makes an important comment regarding the measures of food intake, and I provide a few minor suggestions about including details in your current Method and Discussion sections. Once you have addressed these minor concerns, I would be pleased to accept your manuscript.

We look forward to receiving your revised manuscript.

Kind regards,

Herb Covington, Ph.D.

Academic Editor

PLOS ONE

Journal Requirements:

2. To comply with PLOS ONE submission requirements, in your Methods section, please provide additional information regarding the experiments involving animals and ensure you have included details on (1) methods of sacrifice, and efforts to alleviate suffering.

3. In your Methods, please specify whether the animals were euthanised at the end of the study or if they were housed for future experiments. Please also specify the method of sacrifice used.

Reviewers' comments:

Reviewer's Responses to Questions

**Comments to the Author**

1. Is the manuscript technically sound, and do the data support the conclusions?

Reviewer #1: Yes

Reviewer #2: Yes

2. Has the statistical analysis been performed appropriately and rigorously? 

Reviewer #1: Yes

Reviewer #2: Yes

3. Have the authors made all data underlying the findings in their manuscript fully available?

Reviewer #1: Yes

Reviewer #2: Yes

4. Is the manuscript presented in an intelligible fashion and written in standard English?

Reviewer #1: Yes

Reviewer #2: Yes

5. Review Comments to the Author

Reviewer #1: The authors present a new and unbiased approach to assessing mammalian voluntary ethanol consumption that has revealed identifiable patterns amongst seemingly heterogeneous individual behaviors. The application of finite fixture modeling to these data appears to be statistically appropriate and consistent with other published work utilizing this approach to assess variability in rodent behavior; however, I must include the caveat that I do not have expert-level knowledge of this modeling approach. Regarding the methods and conclusions, I cannot identify any major methodological problems with these studies; the paradigm used is well-established and appears to have been carried out with care and the use of appropriate controls, and the finding that the majority of animals do not escalate their ethanol intake when given unlimited access in a two-bottle choice procedure is consistent with the findings of many others. This study adds to our understanding of ethanol self-administration by identifying two distinct drinking trajectories amongst the minority of mice that do increase their ethanol intake over time, and future studies could apply this approach to determine whether their are behavioral or physiological differences in the "early" vs "late" drinkers, either at baseline or following adolescent ethanol access.

I do have two comments, one minor and one of moderate importance. Firstly, in Figure 4, food intake is only reported through day 38; given that Group 1 mice do not begin to escalate their ethanol intake until sometime after day 34, it could be that food intake is altered in this group once their ethanol self-administration increases. Did the authors measure food intake in these mice after day 41? Could their food intake have changed as their consumption increased?

My minor comment is that in section 2.3 there appears to be an accidental insertion of a partial citation (Gordon & Axelrod).

Reviewer #2: I find this study by Yu and colleagues to be genuinely thoughtful and carefully executed. I only have a small number of comments to help bolster the impact of the results and meaning of the data in the context of alcohol drinking studies using various approaches in mice.

1. In the Discussion, I suggest including a note about the very first significantly notable difference that emerges between the groups of mice indicated by the modeling approach. Is there a meaningful or sensitive way to predict the onset of those individuals most vulnerable to patterns of high intake? Would that ability to detect differences between vulnerable or resilient populations be valuable to future characterizations of molecular and cellular endpoints in adolescent mice?

2. Also with regards to the Discussion more detail should be added about: (1.) the levels of alcohol intake, including a note about blood alcohol levels across the three groups of mice derived from the modeling approach, and (2.) how the levels (g/kg/day) of drinking in this study compare to levels consumed in similar and alternative/binge drinking studies (e.g., Matson and Grahame 2011-Addiction Biology; Thiele and Navarro 2015-Alcohol).

3. In the Method Section, Section 2.2, please provide more details about the when the mice in this study received the two different concentrations of alcohol (i.e., 5% and 10%) and the frequency of each concentration's presentation over the course of alcohol access.

6. PLOS authors have the option to publish the peer review history of their article (what does this mean?). If published, this will include your full peer review and any attached files.

Reviewer #1: No

Reviewer #2: No

---

## [Author Response · Author response to Decision Letter 0]

1 Mar 2025

We have addressed every point of the reviews, and a file named 'Response to Reviewers' is included.

---

## [Editor Report · Decision Letter 1]

7 Mar 2025

Developmental trajectory of voluntary alcohol consumption in adolescent mice using finite mixture  modeling and Bayesian posterior probability analysis

PONE-D-24-56621R1

Dear Dr. Yu,

We’re pleased to inform you that your manuscript has been judged scientifically suitable for publication and will be formally accepted for publication once it meets all outstanding technical requirements.

Kind regards,

Herb Covington, Ph.D.

Academic Editor

PLOS ONE
---

## [Editor Report · Acceptance letter]

PONE-D-24-56621R1

PLOS ONE

Dear Dr. Yu,

I'm pleased to inform you that your manuscript has been deemed suitable for publication in PLOS ONE. Congratulations! Your manuscript is now being handed over to our production team.

Kind regards,

on behalf of

Dr. Herb Covington

Academic Editor

PLOS ONE